# Toward Building a Functional Image of the Design Object in CAD

**Vladimir Shevel, Dmitriy Kritskiy *** and **Oleksii Popov**

Department of Information Technology Design, National Aerospace University "Kharkiv Aviation Institute", 61070 Kharkiv, Ukraine

* Correspondence: d.krickiy@khai.edu

**Abstract:** The paper proposes an approach to the classification of lifecycle support automation systems for engineering objects, with the proposed structure of the description of the designed object, using a triple description approach: functional, mathematical, and physical. Following this approach, an algorithm for drawing up a functional description of the lifecycle is described, which is based on the principle of unity of analysis and synthesis of the created system in the design process. The proposed solutions are considered using the traditional aircraft shaping methodology with the application of the airplane make-up algorithm as an example. Furthermore, the architecture of a multiagent platform for structural–parametric synthesis of the object was presented; for convenient usage of this architecture, it was proposed to use classification of design tasks in the form of a design cube. The proposed approach allows obtaining an accurate description of the designed object and the subtasks needed to create it, which can reduce the time of the project. Unfortunately, not all decisions can be automated at the given stage of technical development, but what is possible to automate is enough to achieve a reduction in terms of realization and an acceleration of the prototyping process, as shown in the considered example. The actual reduction throughout the lifecycle of the product ranges from 10% to 21% of the planned time.

**Keywords:** lifecycle; CAD; engineering design; multiagent platform

## 1. Introduction

The desire to automate as fully as possible the support of the lifecycle of an engineering object (LCEO) has led to the creation of numerous examples of automated systems in the form of application packages and variants of their combination in the form of integrated computer systems (ICS). Most publications on the subject of lifecycle management are aimed at proving that the formalization of the design process is necessary, in order to achieve a reduction in both implementation time and cost. For example, the authors of [1] showed the need to create a formalized method for obtaining comparable and transparent assessments of lifecycle sustainability indicators in the early stages of design and planning of a civil works project, which allows comparing the sustainability indicators of design concepts at the lifecycle stage and at the level of building components.

The latter position themselves as some "PLM solutions" of a comprehensive nature, built using continuous acquisition and lifecycle support (CALS) technologies on the basis of the unified information space (UIS) of the LCEO. For example, in [2], the LCEO was described using the example of aviation technology to achieve the minimization of harmful emissions throughout the entire lifecycle, while describing several tools that complement each other, but duplicating functionality was also shown.

The authors of [3] demonstrated the diversity of computer-aided design (CAD) systems in modeling, and they also proposed the function and geometry exploration (FGE) approach, which allows designing on the basis of the description of functionality and geometry. As noted in the publication, this is very good for prototyping (obtaining a large

number of different options); however, in general, it is not enough for the implementation of lifecycle management at all stages.

The main disadvantage of such tools is the duplication of a number of functions.

The authors of [4] classified and analyzed various lifecycle support models, revealing that existing systems complement, duplicate, and, in some cases, have redundant functionality. These models focus on a continuous development strategy, where the information gathered during the cycles can serve as useful input for managing future projects or even expanding and improving the current project, while taking into account possible risks.

In particular, practically all systems of the considered purpose support elements of design automation with respect to various engineering objects. This is due to the fact that design elements take place when solving almost all tasks of supporting the lifecycle of an engineering object (LCEO). Examples of such systems are the following [5]:

CAD (computer-aided design). They automate the procedures of geometric modeling and design documentation generation in an engineering object design process. A set of geometric models and design documentation is considered as a design object.

CAE (computer-aided engineering). These are automation systems for engineering analysis, which is seen as an element of engineering design. Modern CAE systems are used together with CAD systems, often integrated into them. Physical and mathematical models of the analyzed engineering object are considered as a design object.

CAPP (computer-aided process planning). The object of this design is the technological processes of the production of objects.

CAM (computer-aided manufacturing). Systems for the computer-aided preparation of control programs automate the procedure of control program design. The object of the design is considered to be the software controlling the technological equipment.

CASE (computer-aided system/software engineering). This is a system for automating the development of information systems or software. Software components are considered as the object of design.

CAQ (computer-aided quality control). The object of computer-aided quality control systems is a quality control system for manufactured products.

FMS (flexible manufacturing system). The object of the design is equipment configurations adapted to the production of specific products and their volume.

SCADA (supervisory control and data acquisition). Data acquisition technology for supervisory control of a process is considered as the design object.

CIM (computer-integrated manufacturing). The object of the design is the combination of different production facilities to produce a particular product.

MRP (manufacturing resource planning). The object of the design is a system of plans used in the organization of production.

MRO (maintenance repair and overhaul). The system of documents regulating the rules of operation and maintenance of facilities is considered as the object of design.

ICS ProEngineer (WildFire), Unigrapphics (NX), and CATIA are integrated computer systems to support LCEO, incorporating the functions of the above packages.

The list of examples could go on.

## 2. Materials and Methods

In this situation, it seems relevant to clarify the concept and content of design, since they are presently individual in nature and depend on the tastes of the developers of automation systems. The solution of this problem makes it possible to unify the approach to automating the design procedure, which creates prerequisites for reducing the duplication of design automation tools in the components of the automated LCEO support systems and in the development of ICS.

The basis for defining the inherent functions of any automation system to be created is to analyze the automation object. In this paper, an attempt is made to identify the functions inherent in CAD system for an engineering object.

In order to solve this problem, it is first necessary to define the concept of design.

Design is one of the most general and "fuzzy" concepts used in engineering practice. The analysis of normative documents related to design [6] does not give an unambiguous answer to this question. Most often, design is indirectly defined through concepts such as "design decision", "design document", "design operation", and "design procedure". In our opinion, the most constructive approach is to define design as a procedure for constructing a complete description of the engineering object to be created.

It is proposed to present a complete description of an engineering object in the form of three components (Figure 1): functional description, mathematical description, and physical description, corresponding to the notions of description of complex objects (systems) contained in [7],

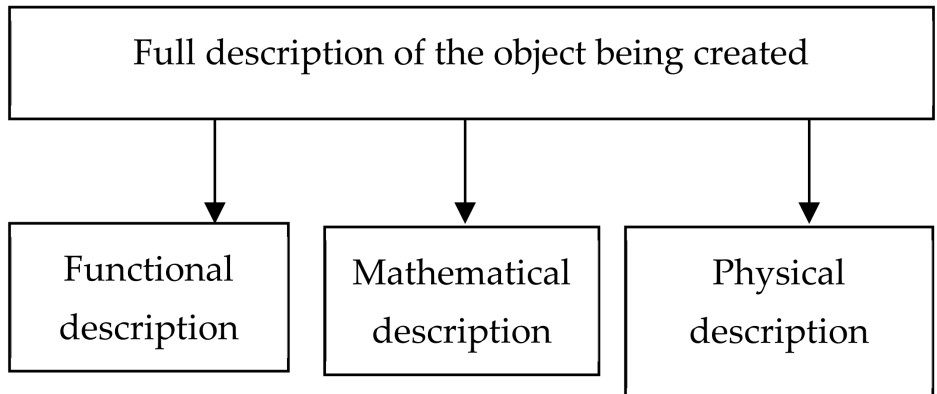

**Figure 1.** Structure of the complete description of the project.

The functional description or F-image of the object to be created is a necessary and sufficient set of functions performed by the object in its functioning as a system of purposeful action. The basis for selecting a set of functions is the general (integral) function of the designed object. It is subjected to successive analysis (detailed elaboration and splitting) in the design process.

The mathematical (algorithmic) description or M-image of the created object is the result of algorithmization of implementation of F-image components. Depending on the specific algorithms, the functions can be implemented in automatic mode using hardware or software, in automated mode, when human participation in their implementation is required, and in "manual" mode, when the allocated function is performed directly by a human.

The physical (real-life) description or N-image is a description of the structural elements of the designed object intended for the in situ realization of the F-image components, functioning in accordance with the M-image components.

Figure 2 shows a simple example of how to describe the design object. Example 1 is a bridge truss element, and Example 2 is a numerical integration program.

The design procedure as a process of constructing F-, M-, and N-images of an engineering object corresponds to the principle of unity of analysis and synthesis in the course of solving a complex problem (Figure 3).

The design procedure is presented as a sequence of procedures for analyzing the functional purpose of an object, formalizing the resulting set of functions, naturalizing the functions as elements of the design of the object, and then synthesizing the design as a whole.

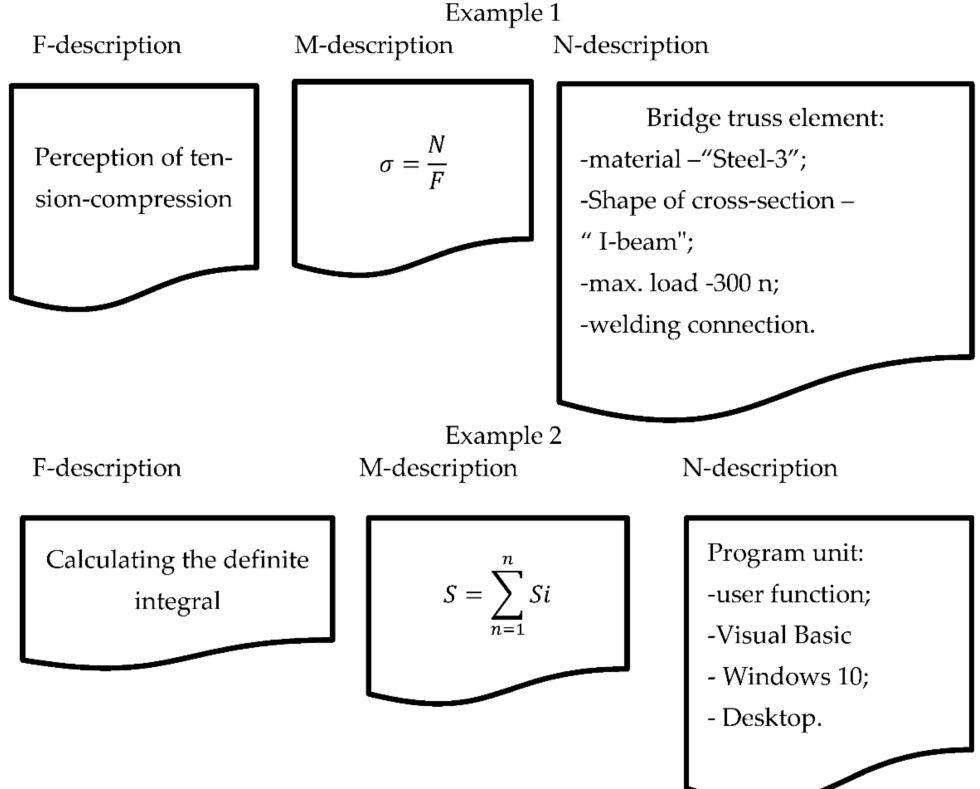

**Figure 2.** Examples of a complete description.

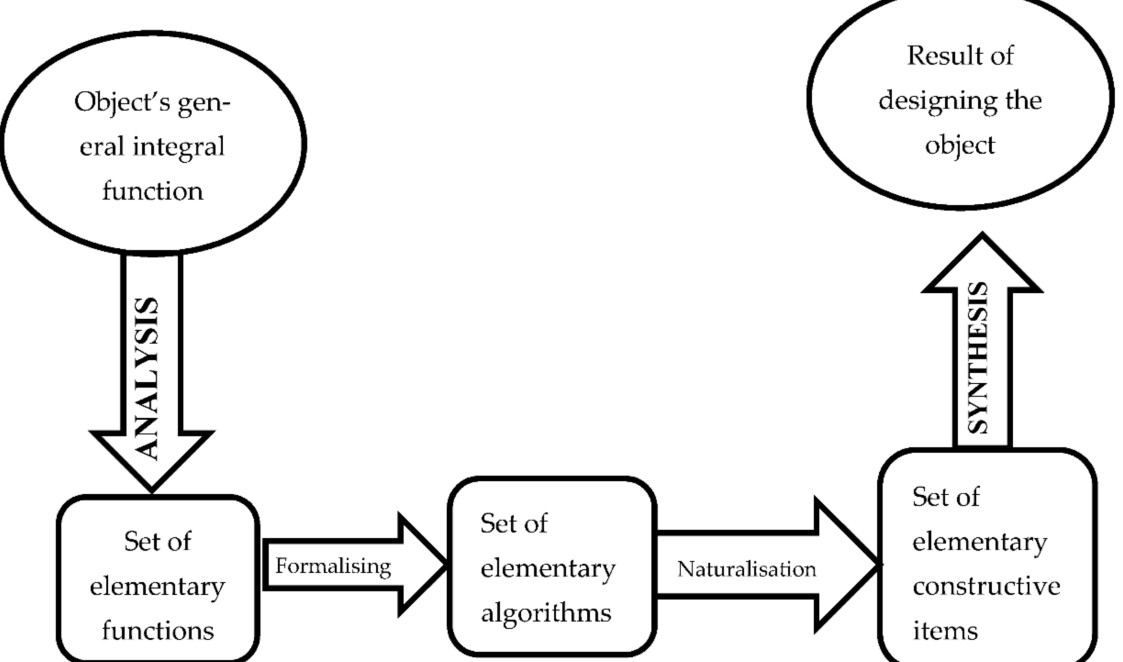

**Figure 3.** Design as a unity of analysis and synthesis.

Naturally, the proposed scheme may contain feedback, indicating the iterative nature of the design process. Iterations are possible at any stage. The following specifics can be considered:

1. The extracted function cannot be formalized to the required level, and a correction of the overall function analysis is necessary.
2. The resulting mathematical description does not provide a solution to the mathematical problem and needs to be simplified.
3. The resulting algorithm cannot be physically realized within known physical principles and requires correction of the algorithm or additional research to develop new physical principles.
4. The resulting set of design elements cannot be combined into a coherent whole due to existing design or technological constraints.

The number of rollback steps can be more than one.

The specifics of constructing the leading component of the description of the projected object are considered the F-description.

The basis for initiating the design is the summary of the requirements for the design object to be created and its overall (integral) function, which are the result of the predesign procedures.

The result of its subsequent analysis, which determines the completeness of the F-image of the designed object, depends on the correct formulation of the general function. The general function is formulated on the basis of a thorough analysis of the requirements statement for the object to be created. It is beyond the scope of this paper to formulate the set of requirements at the level of pre-project procedures. It should only be noted that the procedure for creating the requirements statement is well developed and can be automated. This is facilitated by the existence of standards for requirements statements, e.g., ISO/IEC/IEEE 29148:2011 for software development. Approaches have been developed and software templates exist to automate the development of a requirements statement.

The correctness of the formulation of the general function can be controlled by having a formal description of it. For example, in [8], it was proposed to use the notion of the main utility function of the system P in the form of

$$P = (D, G, H), \tag{1}$$

where D is the action carried out in the execution of the function, G is the object on which the action is being performed, and H is the condition under which the action takes place.

The proposed description can be extended by specifying the nature of the data transformation performed during the action. For this purpose, for example, in [9], the notion of a technical function of the system F was introduced as

$$F = (P, Q), \tag{2}$$

where

$$Q = I \rightarrow O. \tag{3}$$

In this case, Q defines the nature of the transformation of some set I consisting of n input operands into a set O consisting of m output operands.

$$\{I_1, I_2, I_3, \ldots, I_n\} \rightarrow \{O_1, O_2, O_3, \ldots, O_m\}. \tag{4}$$

Such a refinement opens up the possibility of formally controlling the correctness of the F-image of the designed object.

The process of analyzing a general function can be represented as its sequential "splitting" of the function F into subfunctions (Figure 4). The authors of [10] showed how many combinations of different solutions can arise in the process of designing samples of complex equipment.

In this way, the analysis splits the overall function F of the created object into a number of simple subfunctions.

$$F \to \{F_{ijk}\}, \tag{5}$$

where i denotes the level of detail, j is the number of subfunctions in the level, and k is the number of subfunctions in the group.

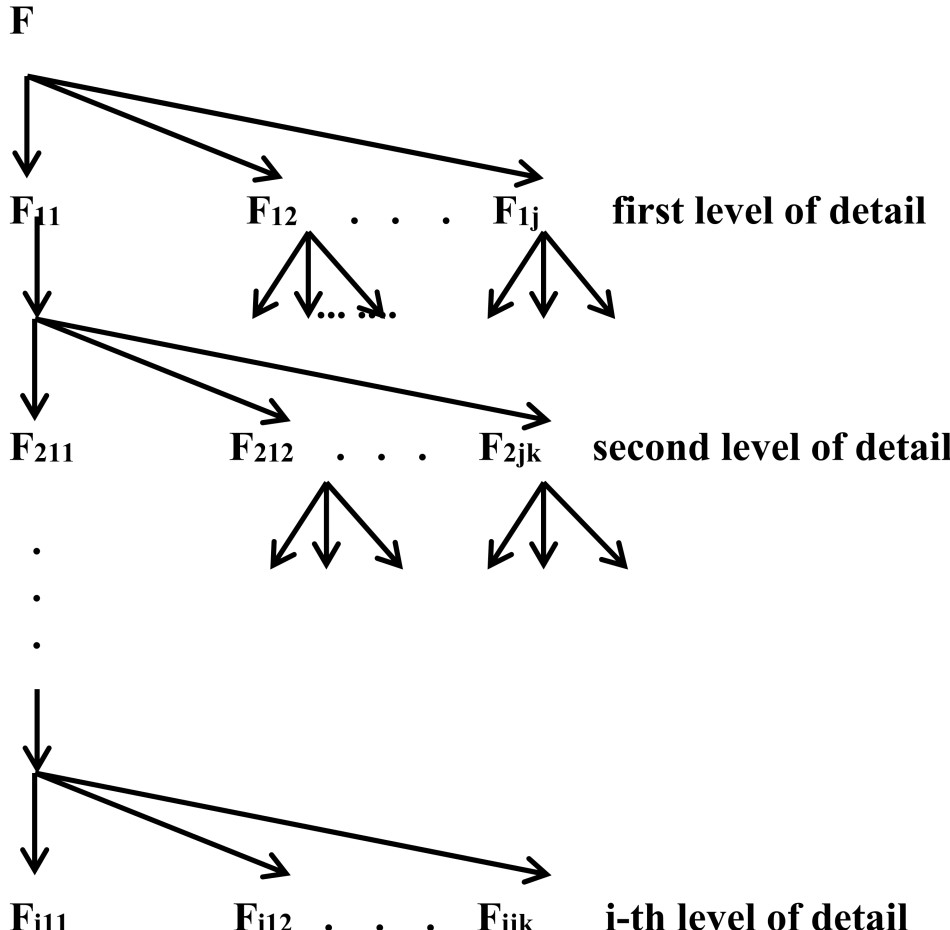

**Figure 4.** Analysis tree.

The analysis results in a set of elementary subfunctions that are not subject to further detail.

The elementarity property can be achieved for different subfunctions at different levels of detail. To determine the objective properties of an elementary subfunction (completion of the itemization process), the following rules can be used:

- A subfunction is elementary if its subsequent algorithmization is not difficult for the designer;
- A subfunction is elementary if the designer knows a variant of its in situ implementation that can be used as a readymade part of the object being created.

Due to the subjectivity of the analysis procedure, the result (a set of elementary subfunctions) is highly dependent on the skills and training of the designer, and it is highly questionable in terms of its optimality.

During the analysis of a generalized function, the designer must constantly monitor the correctness of the detailing, which consists of checking the implementation of the analyzed function by means of the set of subfunctions resulting from its detailing. For this purpose, it is sufficient to represent each subfunction in the form of Equation (4), and then analyze the set of subfunctions for joint implementation. In [11], we proposed a procedure of such analysis, which consists of "extinguishing" the internal operands and comparing

the obtained result with the description of the analyzed function in the form of Equation (4). The absence of necessary operands or appearance of new ones, which are not used in the description of the detailed function, indicates incorrect specification or an error in assigning operands to subfunctions.

The efficiency of constructing an F-image of the projected object can be improved by using readymade variants of detail at the upper levels during the analysis. Indeed, experience shows that several initial levels of detail can be the same for similar objects. For example, in the design of most software systems, a generalized function at the first level of detail is represented as three invariant components:

1. Data entry;
2. Obtaining a result;
3. Output of the result.

This fact is reflected in HIPO diagrams [12], popular among software developers in the early stages of software design technology.

The generalization of experience in designing various objects will make it possible to identify invariant levels of F-image description for similar objects. This will significantly improve the quality of design, as the invariant levels define the design at the initial stage, where miscalculations are most likely to have serious consequences for the project as a whole.

In the course of designing a new object, the designer may encounter functions that are being implemented for the first time in engineering practice. For such functions, there may not be a corresponding physical principle of operation (PFA). To solve such problems in CAD, there should be special means of checking the existence of the physical principle of action and, if necessary, the synthesis of a new physical principle of action on the basis of known physical and technical effects (PTE) [13]. In the presence of the function description in the form of Equation (4), as well as a library of descriptions of existing PTEs, the task can be solved in an automated mode in accordance with the scheme in Figure 5.

As can be seen from the block diagram in Figure 5, the required PFA is synthesized as a "chain" of PTEs, which are described in some library. The "chains" are formed on the basis of correspondence of O-operands and I-operands of the considered PTEs. Several competing "chains" can be obtained as a result of solving the problem. As a criterion for choosing the best solution variant, one may use, for example, time, cost of implementation of a new PTE, and accuracy of the result obtained when using it in the course of design.

All newly synthesized FPPs are accumulated in the type library and can be used in the future.

To automate the task of synthesizing a new PFA, off-the-shelf tools can be used, examples of which are software packages such as Relko (Hilden, Germany), "Inventing machine" (Minsk, Belarus), Tech Optimizer (Oregon City, OR, USA), Product Function Definition (Washington, DC, USA), and Product Function Optimizer (Washington, DC, USA).

Concluding the overview of the procedure of building the F-image of the designed object, it should be noted that the analysis procedure is insufficiently formalizable; in modern CAD, it is performed in the "manual" mode. It can be improved by developing information support tools for the designer. In particular, it is advisable to use automated reference information about the results of designing similar or similar objects.

Procedures for controlling the correctness of the results of analysis and synthesis of new FTPs, although not present in modern CAD, are sufficiently well formalized and can be implemented in new designs.

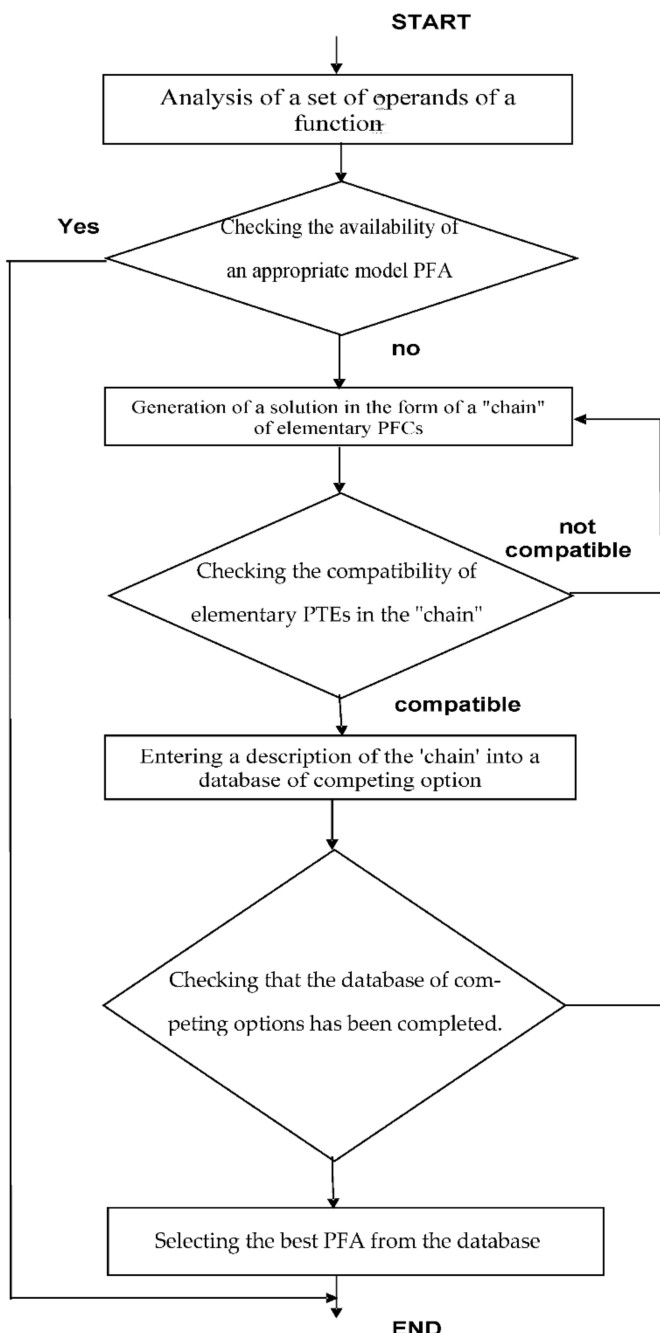

**Figure 5.** Synthesis scheme.

### 3. Results

One of the main provisions of the aircraft design methodology is to single out two levels of project development—external and internal design [13]. The fundamental difference between them is due to the difference in the final goals of the product development processes implemented here. The purpose of external design is to determine the feasibility and feasibility of creating a product, and the purpose of internal design is to obtain information necessary and sufficient to create a product under specified conditions. The internal design process begins with the overall concept of the aircraft and the shaping of its appearance. It is the task of shaping the appearance that serves as a connecting link in organizing the interaction of external and internal design systems, as shown in Figure 6.

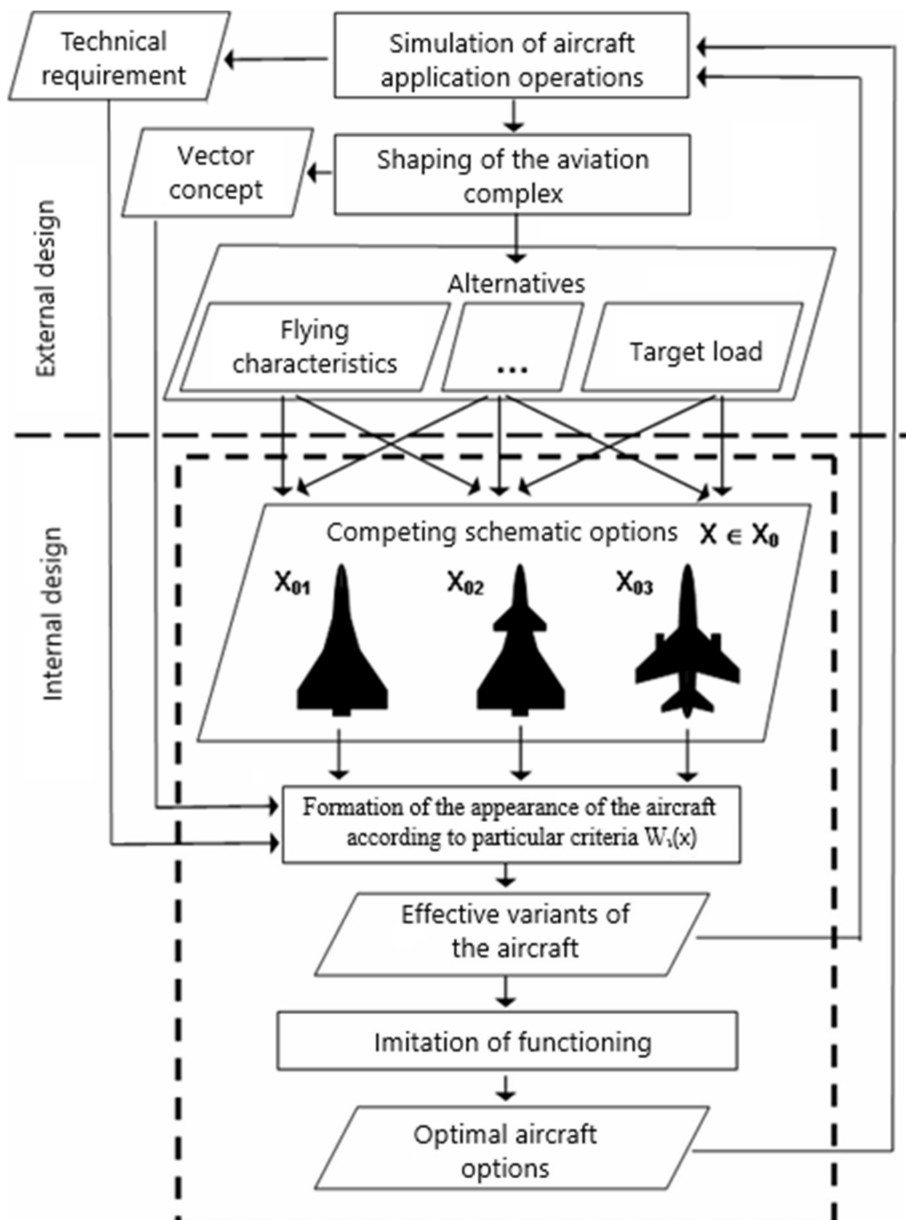

**Figure 6.** Model of interaction of external and internal design systems.

### 3.1. Aircraft Shaping: Traditional Methodology

The concept of "product appearance" does not have a strict definition, but most design specialists include structural (schematic) features and the most important parameters of the design object in its composition, which uniquely determine its shape, size, and takeoff weight. Schematic features include the aerodynamic scheme ("normal", "duck", or "tailless"), the location of the wing (lower, middle, or upper), the shape of the wing in plan, types of wing mechanization devices along the leading and trailing edges, the scheme plumage (low-lying or T-shaped), and type and location of engines. This group of features is called shape parameters. They define a "dimensionless" prototype of the aircraft, the dimensions of which are further determined by subsequent calculations. Within the framework of the selected combination of schematic features, the dimensions of the aircraft are determined primarily by the wing area and the starting thrust of the engines (or the specific load on the wing and the starting thrust-to-weight ratio derived from them). This group of parameters is referred to as dimension parameters.

The perfection of the chosen scheme is characterized by functions of the airframe geometry such as the lift coefficient Cyα, drag coefficient Cxα, aerodynamic quality K, and the efficiency of the power unit, which are functions of the gas dynamic characteristics of the engine, e.g., the specific fuel consumption Cp and the relative fuel mass mt [14].

Together, these parameters uniquely determine the flight characteristics of the aircraft. By varying them, the designer achieves the design goals.

Thus, the problem of shaping the appearance (AS) includes the subtasks of structural synthesis (determination of circuit solutions) and parametric synthesis (determination of the optimal values of the dimension parameters).

In this case, the problem of structural synthesis cannot be finally solved in isolation, since the efficiency of the obtained circuit solution can be confirmed only as a result of solving the problem of parametric synthesis.

It is this circumstance that makes the AS task so important and so difficult to formalize. The complexity of this task is not so much in organizing the enumeration of permissible combinations of circuit features, but in the difficulties of comparing synthesized variants of circuits.

The complexity of the considered problems is highlighted in Figure 6 with a dot-and-dash frame. At the same time, this work does not consider the formation of a geometric model of an aircraft; it can be built interactively or generated automatically in a specialized knowledge-oriented CAD subsystem, which does not affect its subsequent use in any way.

The traditional solution to the problem of selecting rational options for design solutions at the preliminary design stage is to use empirical or approximate analytical dependencies to determine the aerodynamic and flight characteristics [14]. The same path was proposed in the works of recent years [15,16]. A similar approach to the AS problem (search for analogs, enumeration of combinations of circuit features, and calculation of characteristics based on simplified dependencies) also prevailed in [17,18].

The motive for refusal from numerical research at the stage under consideration is usually its high labor intensity. For all the seeming naturalness of this approach (many initial data for accurate analysis have simply not been established yet), it has a serious drawback—the low accuracy of estimates of the aircraft's functional properties, i.e., its aerodynamic and flight characteristics.

At the same time, only flight characteristics can act as private criteria for evaluating and comparing options at the preliminary design stage. The main source of regulatory data is the airworthiness standards for aircraft of various categories (e.g., [19]), and practically the only category of aircraft properties that can be checked for compliance with the standards at this stage is flight characteristics, since these properties have the most complete quantitative representation in them. Therefore, it is quite natural to strive to obtain a more accurate assessment of these characteristics at the earliest possible stages of development. However, the well-known simplified dependencies cannot cover the entire wide range of speeds specified in the norms.

The feasibility of such a solution is supported by the leading trends in the development of software engineering analysis (CAE) in recent years, primarily the following:

- The transition to multidisciplinary (so-called multiphysics) modeling for solving related problems requiring the simultaneous analysis of processes of different physical nature, e.g., the gas flow around a structure and its deformation under the action of this flow; such capabilities can be both embodied in one multidisciplinary system, for example, VHDL-AMS simulators, SimulationX, ANSYS multiphysics, and NASTRAN multidisciplinary, or implemented through the interaction of multidisciplinary systems, such as FlowVision and ABAQUS;

- Transition from design verification to up-front simulation; the experience of using the SimDesigner 2004 toolkit in the CATIA V5 environment showed that transferring the analysis of design solutions to earlier design stages significantly increases the efficiency of CAD; the development of this direction was the SIMULIA project by Dassault Systemes, which laid the foundation for a new class of CAD tools—realistic

simulation, also called Rapid Analysis and Validation of Design Alternatives (RAVDA); similar developments were carried out by other companies;

- The use of CAE systems not only for verification, but also for the synthesis of design solutions; typical examples are the methods of optimization of the shape of objects (topology optimization) by iterative execution of procedures for the analysis of the stress state and subsequent exclusion from the model of the least loaded finite elements, as a result of which a structure close to equal strength is formed.

An additional incentive to include software tools for the analysis of aerodynamic processes, computational fluid dynamics (CFD), and flight simulation systems together with specialized programs for synthesizing the layout and geometry of the model in a single cycle of forming the appearance of the aircraft is another modern trend in the construction of CAD—the transition from "initially integrated" complexes to "freely integrated" sets of functional modules. This approach (arrangement of readymade modules with a minimum of own programming) is successfully used in the design of microelectronic and micromechanical products, where such systems are called heterogeneous CAD systems. This path seems to be promising for other areas of technology, which is confirmed by the developments of foreign research groups, particularly the work of the Delft Technical University [20].

Thus, the analysis of previous works indicates the existence of a contradiction between the need for multivariate design, the requirements for reducing the development time, and the insufficient accuracy of methods for assessing the aerodynamic and flight characteristics of an aircraft during preliminary design. To resolve this, it is proposed to use CFD systems and flight simulators at the earliest possible design stages from the moment of synthesis of the first variants of the topology and geometry of the aircraft, i.e., already in the problem of shaping the appearance.

The general sequence of design procedures in the proposed preliminary design methodology as a whole also corresponds to Figure 6; however, their content changes as follows:

- Selection of a combination of schematic features for the next design option:
- Design calculations of the main parameters of the aircraft;
- Building a geometric model of the first approximation;
- Virtual blowdown of the model in the CFD system;
- Determination of the main aerodynamic characteristics;
- Setting the initial data and/or programming the flight dynamics block of the flight simulator;
- Virtual test flights in the simulator;
- Conclusion about the possibility of achieving the specified characteristics and about compliance with airworthiness standards.

To test the proposed preliminary design technology, a prototype of the software package was developed as part of the SolidWorks CAD system, the Plane3D proprietary application for the automated synthesis of the geometric model, the Flow Vision CFD system, and the Flight Gear flight simulator. The complex also uses general-purpose software—the Notepad text editor for storing the coordinates of the points of standard aerodynamic profiles, the MS Excel spreadsheet processor for storing the characteristics of analog aircraft and calculation results, and the Blender graphic editor for converting the 3D model into the format required by the flight simulator. The scheme of the complex is shown in Figure 7.

Below, the main stages of work on the analysis of the flight characteristics of the designed aircraft are illustrated.

The geometric model of the aircraft, generated by the Plane3D application in cooperation with SolidWorks (Figure 8), is translated and transmitted to the Flow Vision CFD system in VRML format.

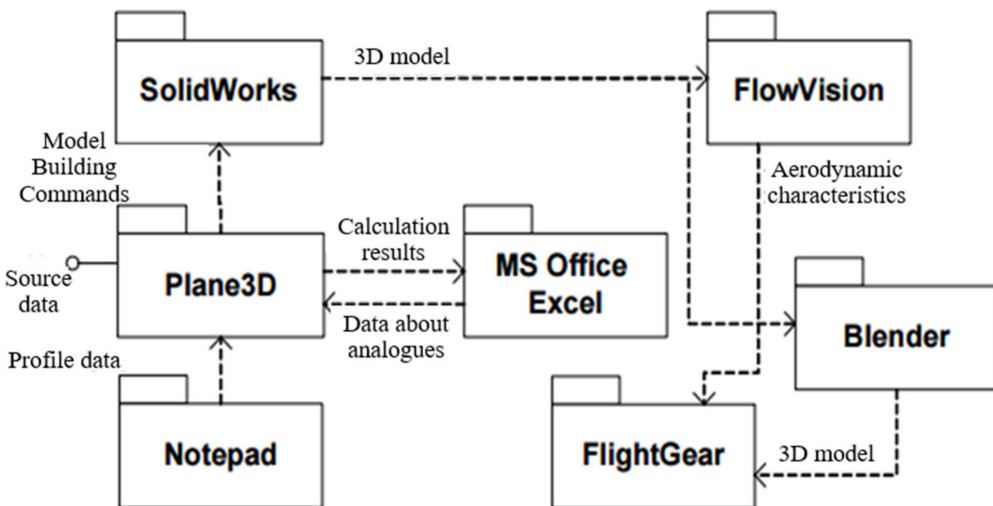

**Figure 7.** Block diagram of the software package.

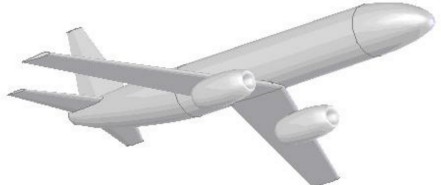

**Figure 8.** Aircraft geometric model.

In the Flow Vision system, a mesh of finite volumes is generated (Figure 9), the conditions for adapting the mesh and rhe boundary and initial conditions are set, and then a virtual blowdown of the model is performed in the mode of interest to the researcher.

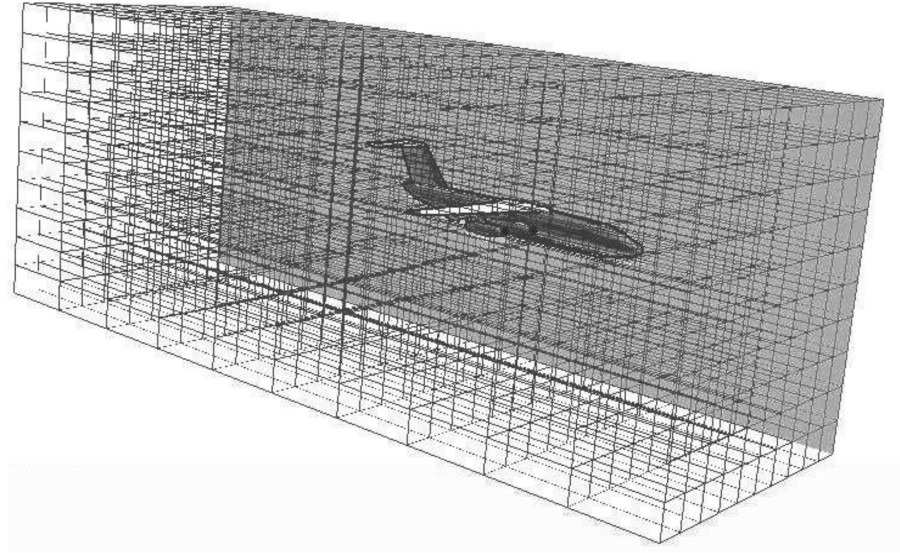

**Figure 9.** Final volume mesh in the Flow Vision system.

The results obtained—values or graphs of changes in flow rates, pressures, aerodynamic forces, and other parameters (Figures 10 and 11)—are subject to processing for the appropriate adjustment of the simulator dynamics block.

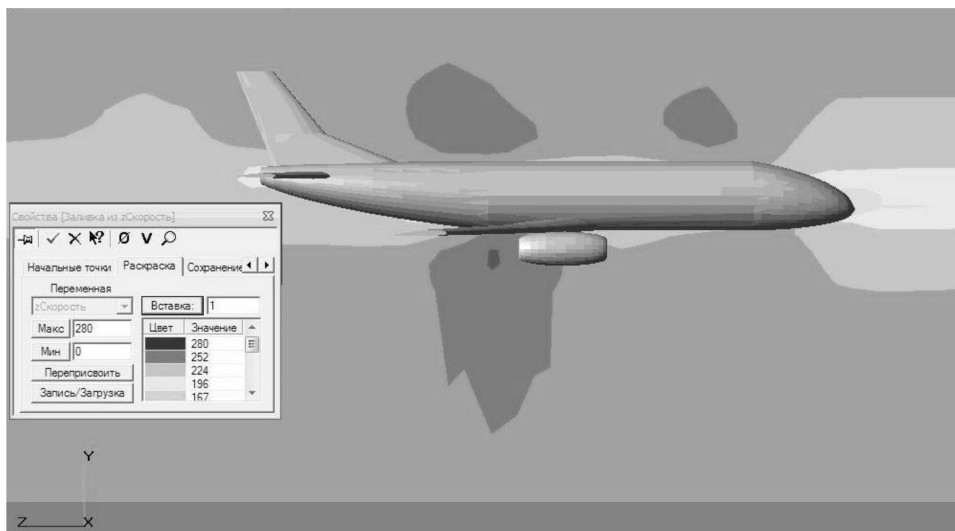

**Figure 10.** Velocity distribution in the plane of symmetry of the aircraft.

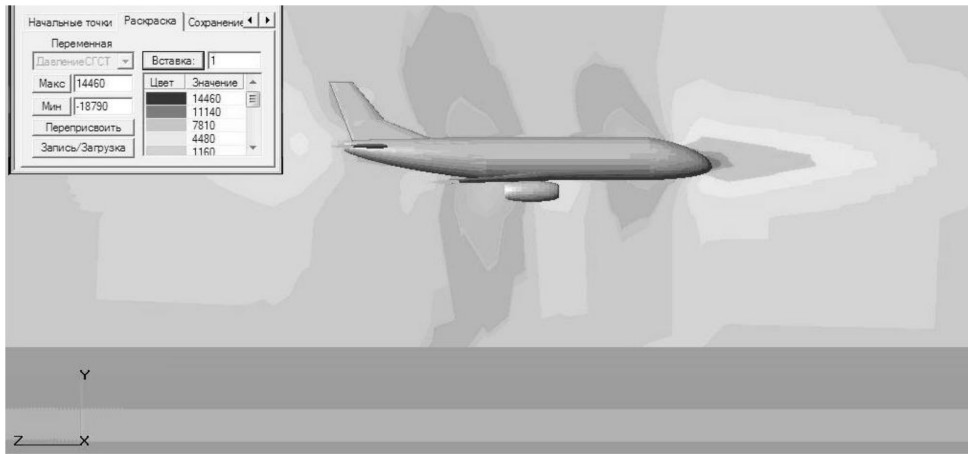

**Figure 11.** Pressure distribution in the plane of symmetry of the aircraft.

After completing the simulator setup, the model is converted to AC3D format and loaded into the simulator to perform virtual flight tests (Figure 12).

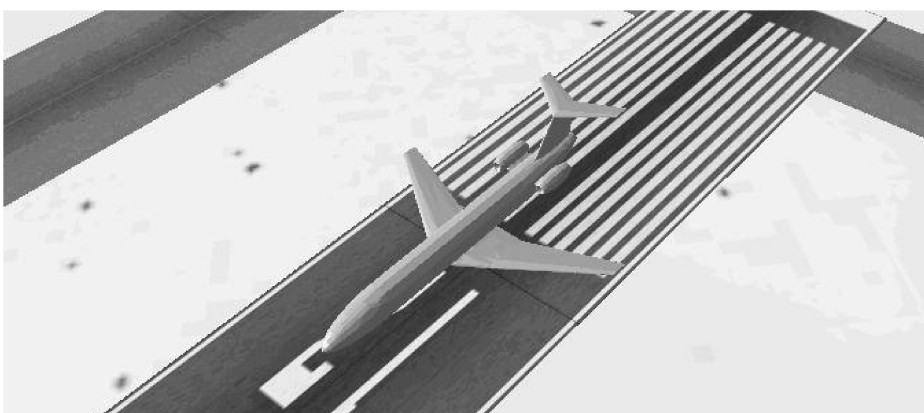

**Figure 12.** Airplane model loaded into the simulator.

*3.2. Architecture of a Multiagent Platform for Structural–Parametric Synthesis of Objects*

The design of complex technical objects requires the simultaneous consideration of a large number of relationships of various types (cause-and-effect, temporal, and spatial)

between their elements and properties and processes of various physical nature (mechanical, electrical, hydraulic, etc.). The construction of a unified informational or mathematical model of such objects is practically impossible, since a different mathematical apparatus is required to describe various types of relations and connections—various methods of continual and discrete mathematics. The decomposition principle leads to the formation, instead of a single model, of a certain set of submodels (private models), each of which reflects a certain aspect of consideration, i.e., a particular point of view on the design object. The number and types of these sub-models can change when moving between hierarchical levels (product, assembly, assembly, and part) and development stages. For an aircraft, the most important of these submodels are geometric, weight, aerodynamic models, models of flight dynamics, power unit, layout and alignment, efficiency, and economic models [14]. Note that the listed models relate only to the functional aspect; in the tasks of resource (strength), structural, and technological design, the corresponding submodels are added to the general list.

However, dividing the description of a complex object into particular models and corresponding groups of tasks significantly simplifies the modeling process within each aspect, while significantly complicating the procedures for corroborating particular design solutions obtained within this framework. Decomposition of a technical system reduces the explicit complexity, but increases the so-called implicit complexity associated with the difficulties in determining the expected properties of the system by the characteristics of its elements, which is a manifestation of the emergence property of complex systems [14].

The entire set of tasks to be solved can be classified according to such criteria as the hierarchical level, aspect, and type of task (see Figure 13).

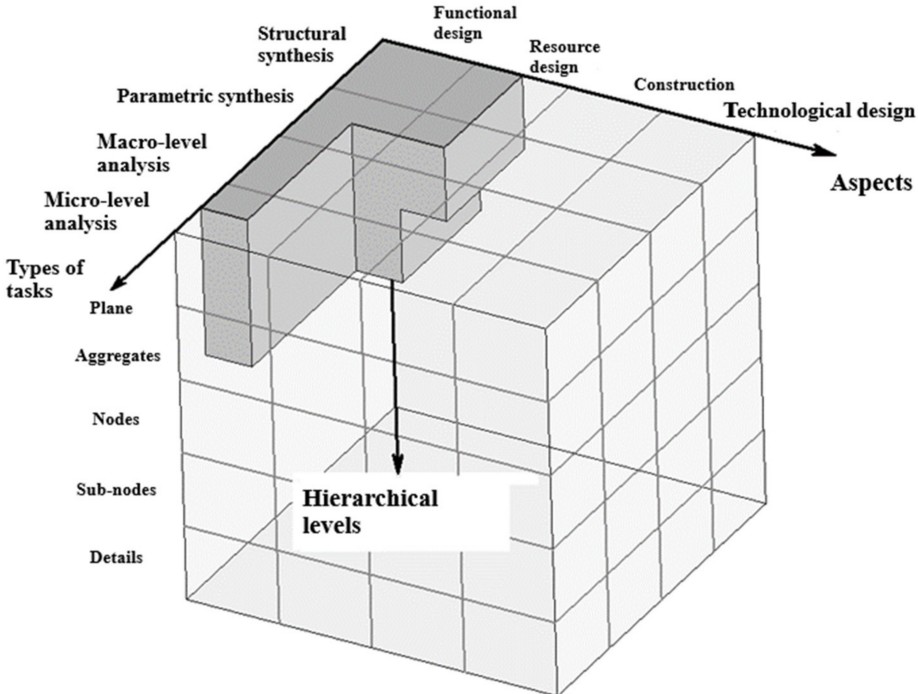

**Figure 13.** Classification of design tasks.

The figure shows an area that roughly corresponds to the tasks of forming the appearance of the product. Table 1 presents a list of design tasks for one of the units (the wing) as an example.

**Table 1.** Aircraft design tasks at the level of the "wing" unit.

| Aspect | Structural Synthesis | Parametric Synthesis | Macro Level Analysis | Micro Level Analysis |
|---|---|---|---|---|
| Functional design | Determination of the wing shape, as well as the types and location of mechanization equipment | Calculation of the geometry of the wing and its parts (area and dimensions) | Calculation of aerodynamic characteristics (Cx, Cy, etc.) | Numerical simulation of the flow around a wing |
| Resource design | Determination of the structural scheme of the wing (types and number of load-bearing elements) | Design calculation of the dimensions of the load-bearing elements | Kinematic and dynamic analysis of moving elements | Numerical simulation of the stress–strain state of the wing elements |
| Construction | Layout of structural elements and equipment (fuel, mechanization drives, chassis, etc.) | Distribution of layout elements by cross-sections | Alignment calculation | |
| Technological design | Choice of the scheme of technological division and assembly <br> The choice of a sheet stamping scheme (drawing, covering, etc.), the number of transitions, etc. | Calculation of assembly dimensional chains <br> Calculation of the dimensions of the workpiece, drawing forces, contact pressure, etc. | Design for assembly <br> Verification calculations of forces, elongation ratio, accuracy, etc. | Numerical modeling of the process of shaping and stress-strain state of the workpiece |

Along each axis of the "system cube" (hierarchical levels, types of tasks, and aspects), movement in only one direction is permissible, but the sequence of individual steps along different axes is not limited by anything other than the availability of the necessary initial data, and nowhere is it stipulated in which direction to perform the first steps, under what conditions to change the direction of movement, and how long it is generally permissible to move in one direction. In addition, we will encounter "linked" (connected) problems, the sets of variables of which intersect, as seen in aeroelasticity problems.

This means that the design strategy may not be as rigidly regulated as with a sequential "aspect" pass. However, this requires a different organization of the information and procedural components of the CAD software. In particular, a flexible management of the sequence of design procedures, close to an adaptive design strategy, can be provided by a multidimensional model of the design object. One of the modern trends in the development of CAD is exactly the desire of developers to build systems of interconnected models that characterize various aspects of describing the design object. An example of such an approach is [14], where the modules of weight design, aerodynamic, and strength calculations were combined. This area also includes the work of a number of foreign research groups, particularly the Delft Technical University (The Netherlands), where the knowledge-oriented multi-model generator (MMG) system was created, the diagram of which is shown in Figure 14. The system uses the facilities of the GDL (General-Purpose Declarative Language). The combined information model of the object covers aerodynamic, strength, and production and economic "layers".

To date, a number of theories have been developed related to solving the problem of multidimensional modeling, the origins of which can be seen in the general design theory of Yoshikawa–Tomiyama, and further traced in numerous modifications of the FBS theory (function—mode of operation—structure) [21]. These theories imply the interaction and origin of one aspect from another, which is absolutely correct in terms of the sequence of design phases. However, for each aspect, the concept of its own knowledge model is introduced and, as a consequence, transitions from one model to another are necessary at each transition to the next design phase.

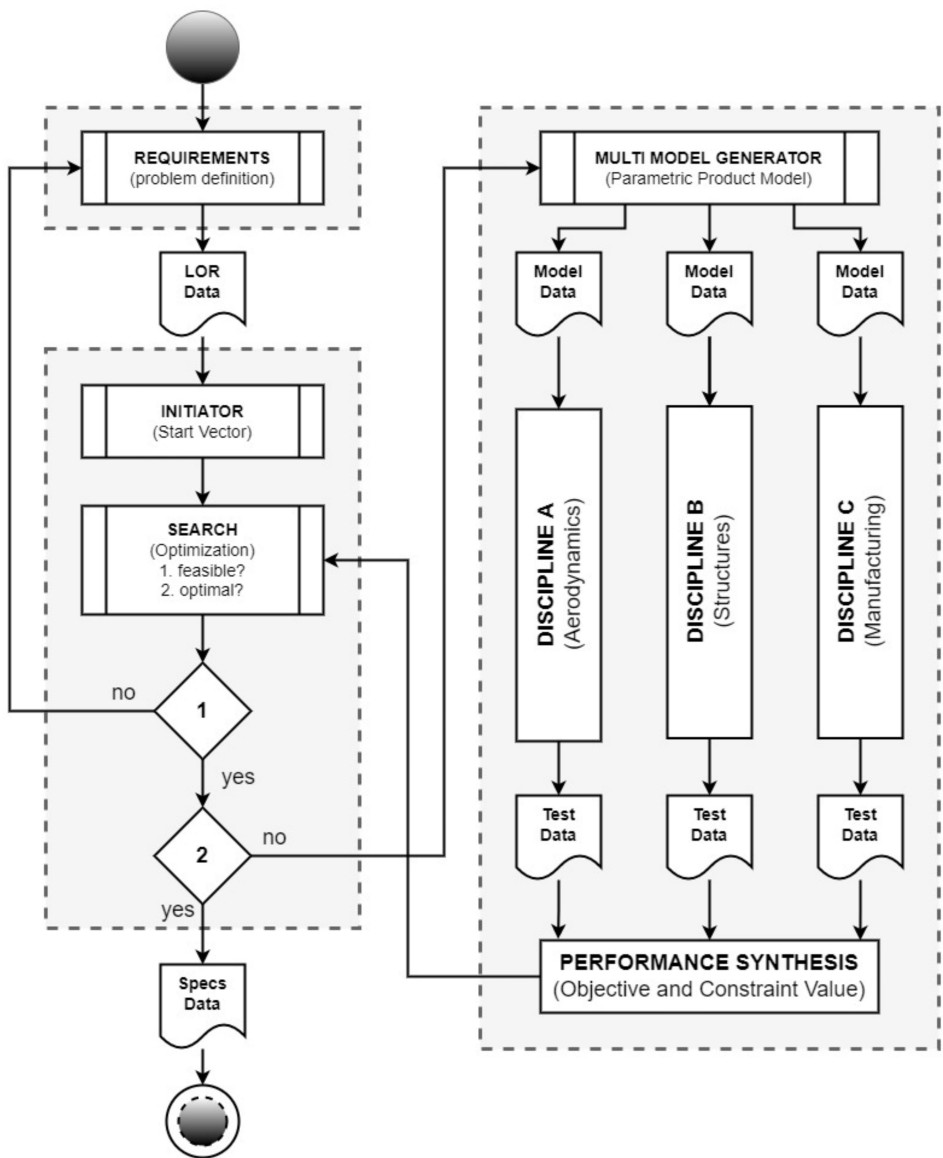

**Figure 14.** Multi-model generator knowledge-oriented system diagram.

For this reason, most theories are considered insufficiently formalized, since the list of knowledge representation models is rigidly defined and cannot be supplemented or changed. In fact, these theories are too formalized, since they formally describe each (but not all) phase and model in advance, which does not allow abstracting from specific design problems. Models and requirements should not be categorized as functional or structural.

If we define a certain format for describing requirements, which can be formalized not within the framework of the specifics of its description, but universally for all possible options for presenting and describing requirements, then the division of design into phases will be extremely indicative, while the work of the system will be carried out outside the design phases.

It can be assumed that the main reason for dividing the design process into phases and stages is, apparently, a person's defensive reaction to two main factors: (a) insufficient amount of human memory, which requires the participation of different specialists to work on one rather complex project; (b) the need to agree on private ("one-aspect") decisions made by different developers outside of contact with each other.

Thus, transferring to the area of computer-aided design all the traditions of organizing the design of the non-automated, a person imposes on the computer their imperfect method of work, generated by the limitations not of the computer, but of its capabilities.

In the design methods of most technical objects, the sequence of stages is set rather rigidly, e.g., for an aircraft: aerodynamics → flight dynamics → structure and strength → technology . . . ; for an electric motor: electromagnetic calculation → thermal calculation → ventilation calculation → noise and vibration calculation → mechanical calculation → reliability calculation, etc.

For an automated system, both of the above factors are not decisive (provided that the procedures for agreeing decisions are also formalized); it really could work "outside the design phases" and there is no need for it to establish the sequence of determining the properties of various groups so strictly. However, this requires completely different design methods that do not imply the division of properties into categories (functional, constructive, costly, etc.) or by physical nature (mechanics, electrodynamics, hydraulics, etc.). It is still difficult to assess the harm from unnecessary restrictions, but it is obvious that everything that constrains the designer's freedom, whether in the content of actions or in their sequence, is not good for the cause.

Thus, it is possible to develop the most flexible system in which the work with different kinds of requirements will be uniform. By delegating calculations and checking compliance with the requirements for specific agents, we get a "design constructor" that any subject matter specialist can use, while accumulating their knowledge in the knowledge base.

Each element of this knowledge base is an agent, which has its own methods of calculation and verification of compliance with requirements. Therefore, when redesigning, the work of the designer is greatly simplified due to the reuse of previously designed elements. At the same time, the properties of an element are not divided into structural and functional. Properties are just parameters of an element, the main function of which is to determine compliance with the requirements; they are only secondarily responsible for visualizing the result—a physical or functional model of the designed product.

From this point of view, various types of design (structural, functional, and conceptual) cease to be decisive factors in choosing a design strategy, including computer-aided design. Multidimensionality cannot be achieved as long as the number of aspects is determined by the programmer developing software for domain specialists. It is necessary not to expand the number of approaches (aspects) to design (this tendency can be seen in a number of works, e.g., the general design theory of Yoshikawa–Tomiyama, i.e., design according to the requirements of the structure, and then the expansion of this theory by the functional aspect of Braha, i.e., paired design [22]), but to develop one universal approach that will define N aspects, where N depends directly on the end user.

The problem of moving from one stage of design to another is an illusion introduced into modern design by the need to recreate the image of the designed product from every point of view, forming a new model for each facet of the design solution. A technologist and a designer, working on one task, see it differently, but the essence of the task does not change from this; if the knowledge of the technologist and the designer were combined, then the design of two models with subsequent coupling and all the resulting difficulties would be meaningless.

Note that, from this point of view, the stage of ensuring the manufacturability of a product design (MPD) between design and technological preparation of production, legalized by many standards, is one of the most odious results of our artificial transition to sequential design; its main content is precisely the revision of part of the design solutions, accepted without taking into account the opinions of technologists (i.e., within one aspect). Here, one can see the loudly condemned "redesign" by all, but raised all over the world to the rank of a mandatory procedure.

The advantage of an automated system is that it can contain an unlimited amount of knowledge in various subject areas, while operating with one model. In this case, it is not necessary, as it is stated in Braha's works, to first check the compliance with the

structural requirements, and then look for intersections with the satisfaction of functional requirements; one only needs to ensure that all requirements are met. Moreover, when solving a particular problem, there will still be a need for additional aspects.

By and large, functional design is just the definition of a list of functional properties (requirements) that the designed object must meet [23]. Even if it is possible to find a solution to the functional puzzle that will fully meet the requirements, there is no guarantee that the structure corresponding to the selected functional solutions will satisfy the structural requirements. Therefore, it is necessary to consider structural, functional, and other requirements as a single entity.

Consequently, the design system should consist of agents with a number of properties, the division of which into any categories is very arbitrary and introduced solely for the convenience of the end user. Each agent has calculation methods and methods for checking compliance with a certain requirement. In addition, the system has a number of requirements; each agent may or may not meet any requirement. However, parametric and structural constraints are also requirements. In such a system, design is reduced to the selection of a set of agents that meet all the requirements. The task of calculating the parameters and structure is a nested selection task, since, when determining the fulfillment or nonfulfillment of the requirement, all possible implementations of the agent are calculated. The sequence of design phases is implicitly taken into account when calculating the model, since previously defined properties are usually used to determine the value of a property. In general, the properties of aspects of earlier design phases are initially determined, e.g., from function to structure.

In accordance with our requirements, an agent is a highly mobile entity that can stop its execution, change its internal structure, and continue execution. The enlarged physical structure of an agent is shown in Figure 15.

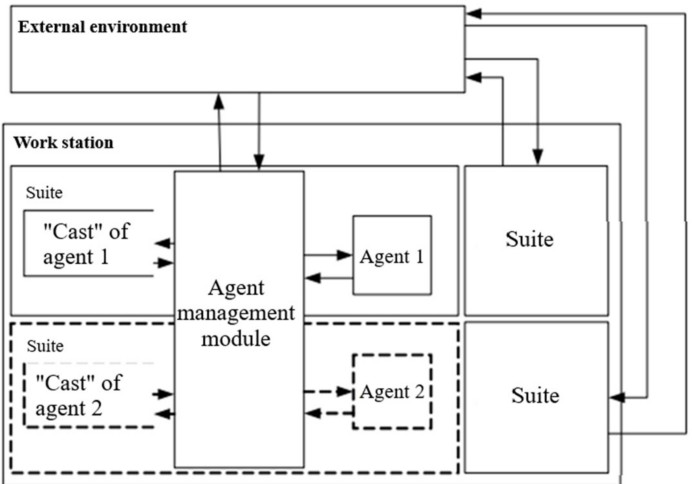

**Figure 15.** Enlarged agent structure.

The definitions of each entity in the figure are provided below.

Let us define an agent as a kind of autonomous entity that is capable of sensing and acting through receptors and effectors. In this case, the software implementation, perception, and impact will be carried out through the transmission of messages. When defining an agent, many people focus on intellectuality. The property of intellectuality depends on the specific implementation of the agent's behavior and does not directly depend on its architecture. The architecture in this case should be at least such that it allows one to show intellectuality. In this case, this can be achieved through input/output channels (sensation/impact).

The external environment is defined as a set of objects that do not belong to a given agent and are perceived as a separate entity. Signals are received from there, which the

agent can ignore if it does not know the kind of signals or respond with some action. Thus, if the external environment is defined as a set of agents, then, for an agent without receptors and effectors, this set will be empty. A larger set indicates that the agent knows more about the environment.

In this case, any agent must have the basic functionality defined in the agent management module; hence, each agent knows about the existence of other agents in its reach. The basic functionality of the agent with the IUnknown interface of the COM architecture can be compared, and then some analogy between the methods defined by this interface can be drawn.

The agent management module is a static part of the agent and is responsible for the implementation of the agent specification in memory, transferring incoming messages and events to it, as well as sending messages to the external environment. It is necessary to stipulate right away that synchronization with other agents, blocking access to the agent, resolving agent links, stopping and starting the agent, and saving its state will take place at this level.

The "cast" of the agent involves the designation of all metadata and dynamic data about an object, i.e., its specification, the last saved state, and associated data.

The agent's suite is the set of data and working modules that relate to a specific agent.

It is necessary to clarify the essence of the relationship between the agent management module and the agent itself. Figure 15 shows the case when one control module can manage several agents. Here, one can draw an analogy with a CORBA server, which, when receiving a new request, sends it to the addressee—a specific instance of an object. In this case, the situation is somewhat different, but the essence remains unchanged; in fact, the agent management module is its server part, which sends messages intended for it to the agent's domain. The creation of a new agent can occur as with cloning the mobile part (e.g., if it is necessary to move a new agent to another runtime, roughly speaking, to another computer) or without cloning the mobile part, where several agents will correspond to one control module. However, below, we take the control module and the agent as a whole. The cloning process of a control module must be transparent to the ultimate creator of the agent. Adopting such an architecture, we get a mobile agent, for which no additional components are needed.

Obviously, even the most experienced specialist cannot handle the representation of integrity when creating complex systems. This is where the myth of the inevitability of dividing the design process into aspects with different models originates. However, an automated system can have a knowledge base of unlimited complexity, and the solution to the problem of choosing an engine for an aircraft will be carried out according to the same algorithm as the choice of a simple mechanism.

The choice of the type of engine can be made on the basis of the dependence of the thrust efficiency on the number M of the flight (Figure 16).

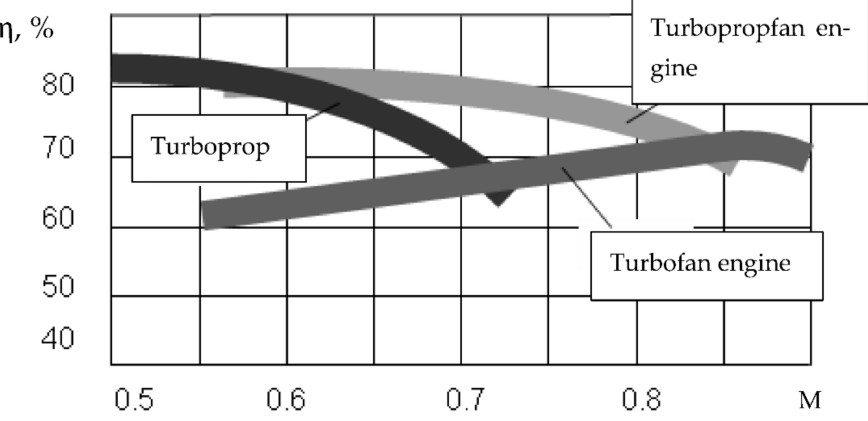

**Figure 16.** Dependence of the thrust efficiency of the engine on the number.

The mechanism of the corresponding module of the knowledge base is the decision table (Table 2). At the same time, the choice of an agent should not be understood as the choice of a finished result from an already existing list.

**Table 2.** Decision table for determining the type and characteristics of the engine.

| Flight Number M | Up to 0.55 | 0.55–0.83 | Over 0.83 |
| --- | --- | --- | --- |
| Engine type | Turboprop | Turboprop fan engine | Turbofan engine |
| ... | ... | ... | ... |
| Diameter of the engine (propeller), m | $(N_0/8.1)^{0.25}$ | ... | $0{,}1{\cdot}G_B{}^{1.5}{\cdot}\varphi_1(R_0){\cdot}\varphi_2(\pi_K{}^*){\cdot}\varphi_3(T_3)$ |

Note: $N_0$—power; $G_B$—air consumption in a second; $R_0$—starting thrust; $\pi_K{}^*$—the degree of pressure increase in the compressor; $T_3$—the gas temperature in front of the turbine; $\varphi_i$—statistical coefficients.

The calculation of the parameters and the determination of the structure in the design is also a kind of choice of values or elements of the product, such as, for example, the calculation of the dimensions of the engine in Table 2. In the case of the multidimensional agent network (knowledge base) described above, the choice of the calculation method is added to these selection parameters, i.e., the choice of a solution in the form of one or another agent.

For an automated system, the traditional division of tactical and technical requirements (TTT) into tactical and technical ones is irrelevant; not only technical requirements, but also a number of constructive ("structural") features are extracted from tactical (in the general engineering sense of "functional") requirements in one iteration, e.g., using production rules of the following form:

*IF* TTTs provide for the standard conditions of use of the aircraft (for example, subsonic passenger or transport), *THEN* the structure of the aircraft should include fuselage, wing, empennage, and landing gear.

*IF* the given cruising speed significantly exceeds the landing speed (determined by the class of the aerodrome specified in the TTT), *THEN* the wing structure should include the flap together with its attachment units.

Hence, it follows that in the conditions of computer support for making design decisions, there is no need to delay the start of the work of designers until the complete appearance of the product, whereas, for technologists, there is no need to delay the start of the work until the preparation of working documentation. The task of modern developers is to overcome the barrier of formalization of knowledge, to present it as a certain entity corresponding to a single, flexible, and scalable pattern.

## 4. Discussion

The considered approach was experimentally used in the development of a small aircraft at the Institute of Physical Modeling Problems of the National Aerospace University "Kharkov Aviation Institute". Table 3 shows the planned indicators in the preparation of the project.

It was revealed that, during the implementation of the project, temporary changes should be made to the organizational structure to reduce the time for coordinating decisions, thereby reducing permits. It is necessary to make the following design changes in the organizational structure of the enterprise for the duration of the project: provide an economist to the project team to transfer the sector of strength, aerodynamics, power plant, and technology to the subordination of the design team, to the subordination of the chief designer the department of avionics and control systems, and to an experimental team, as well as create conditions for the possibility of using pilot production through the chief designer. This does not mean that it is necessary to disband the existing divisions, but an internal order should be created, according to which, for the duration of the project, there will be such an organizational structure of the project itself. The actual data obtained during the implementation of the project are presented in Table 4.

**Table 3.** Planned indicators of the project.

| Stage Name | Stage Duration, Months | Duration in Relation to the Entire Project % |
|---|---|---|
| Marketing research | 1 | 3.33 |
| Rationale for the possibility of creating | 2 | 10.00 |
| Avan project (feasibility study) | 2 | 16.67 |
| Submission of an application to the state aviation agency | 1 | 20.00 |
| Preliminary design | 2 | 26.67 |
| Creation of a layout | 0.25 | 27.50 |
| Creation of a technical product | 2 | 34.17 |
| Creation of working design documentation | 2 | 40.83 |
| Preparing for the production of a prototype | 1 | 44.17 |
| Prototype production | 6 | 64.17 |
| Test preparation | 1 | 67.50 |
| Flight development tests | 6 | 87.50 |
| Correction of design documentation | 0.25 | 88.33 |
| Certification tests | 2 | 95.00 |
| Documentation approval | 1 | 98.33 |
| Obtaining a type certificate | 0.5 | 100 |

**Table 4.** Factual data.

| Stage Name | Stage Duration, Months | The Duration of the Project in Relation to the Planned Dates % |
|---|---|---|
| Marketing research | 1.00 | 3.33 |
| Rationale for the possibility of creating | 2.00 | 10.00 |
| Preliminary project (feasibility study) | 2.00 | 16.67 |
| Submission of an application to the state aviation agency | 1.00 | 20.00 |
| Preliminary design | 5.50 | 38.33 |
| Creation of a layout | 0.25 | 39.17 |
| Creation of a technical product | 1.00 | 42.50 |
| Creation of working design documentation | 1.00 | 45.83 |
| Preparing for the production of a prototype | 1.00 | 49.17 |
| Prototype production | 4.50 | 64.17 |
| Test preparation | 1.00 | 67.50 |
| Flight development tests | 1.25 | 71.67 |
| Correction of design documentation | 0.00 | 71.67 |
| Certification tests | 0.75 | 74.17 |
| Documentation approval | 1.00 | 77.50 |
| Obtaining a type certificate | 0.50 | 79.17 |

It should be noted that products of complex technology are, first of all, unique; therefore, projects for their creation are quite unique in nature.

From the data obtained, it can be seen that the actual data throughout the lifecycle receive a significant acceleration at those stages that are associated with the perception of the initial data and the performance of work without the accompanying errors associated with a misunderstanding of the product requirements. Specialists choose the description they need and perform the tasks assigned to them more efficiently. However, such good results can also be associated with the experience of implementing similar projects among

specialists from each department. For each design bureau, the percentage of time saved will be different, ranging from 7% to 21%.

## 5. Conclusions

The design procedure is invariant for a wide class of engineering objects and represents the construction of a complete description of the designed object, consisting of three components: F-description, M-description, and N-description.

The leading component of the description is the F-description of the object.

The procedure for constructing a functional description is formalizable and can be performed in automated mode. It is advisable to use a standardized procedure in the design of the software to support the LRC. During testing of the prototype of the complex, the main results were obtained, which are given below.

The possibility of including the problems of aerodynamics analysis and flight simulation in a single aircraft design process at the early stages of development was established.

When using the described approach, the development time was reduced to 7% if the development was completed by a preliminary project. In the case of using the proposed approach throughout the entire lifecycle of the project, a reduction of up to 21% was achieved.

The need for a more thorough construction of a geometric model than is usually accepted in the tasks of shaping the appearance is revealed due to the sensitivity of CFD systems to the smoothness of the model surfaces.

The main difficulties that prevent the complete automation of the task of forming the appearance of the aircraft are as follows:

- Lack of objective information about the accuracy of simulation results in various simulators; this may require separate studies to compare the capabilities of different flight simulation systems;
- Absence (for objective reasons) at the early stages of development of a number of initial data for the simulator (these data, in principle, cannot yet be obtained); this may require searching for a prototype by morphological methods (e.g., [18]) and using its characteristics as a first approximation;
- Labor intensity processing the results of aerodynamic analysis and preparing data for the simulator on the basis of the results of virtual blowdown; this may require the creation of translator programs;
- Lack of a unified format for presenting the aerodynamic and flight characteristics of an aircraft.

**Author Contributions:** Conceptualization, V.S. and D.K.; methodology, D.K.; software, O.P.; validation, V.S., D.K. and O.P.; formal analysis, V.S.; investigation, D.K.; resources, O.P.; data curation, O.P.; writing—original draft preparation, D.K.; writing—review and editing, V.S.; visualization, V.S.; supervision, D.K.; project administration, D.K.; funding acquisition, O.P. All authors have read and agreed to the published version of the manuscript.

**Funding:** This research received no external funding.

**Institutional Review Board Statement:** Not applicable.

**Informed Consent Statement:** Not applicable.

**Data Availability Statement:** Not applicable.

**Conflicts of Interest:** The authors declare no conflict of interest.

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
