# Peer review of "Toward Building a Functional Image of the Design Object in CAD"

_computation, doi:10.3390/computation10080134_

Round 1
Reviewer 1 Report
The paper claims the generation of a novel architecture to support the design objective in CAD. It starts with a description of relevant acronyms, a deeper description of the used nomenclature, and then the presentation of the new architecture. The results are presented with an example in the form of an airplane design.
The reviewer does not recommend the publication of the paper in the current form. It should be majorly revised and submitted for an additional review. The reviewer found numerous inconsistencies not just in the form of the paper but also the description of the content. It is recommended to the authors to thoroughly review the manuscript for its comprehensiveness.
Some examples:
- Some words are introduced with acronyms and later still written out (e.g., CAD), ans other words are not introduced but show only the acronym (e.g., CALS).
- Figure 2: Sigma is not N via F except N is supposed to be a force and F is an area. If so, it should be noted as it is unconventional.
- VHDL-AMS is a hardware description language and works on a different level than, e.g., ANSYS Multiphysics
- CFD means Computational Fluid Dynamics, it is introduced as aerodynamic processes (the reviewer of course understands the author's intentions but it is misleading to the untrained reader)
- in the conclusion the problem of geometry - CFD is mentioned, but nowhere themasized in the paper.
Additionally:
- the language is unprecise at places, e.g., line 382: changes somewhat. This is not a scientific description of the changes.
- Figure 9: It looks like the mesh is not refined close to the airplane surface. How is this problem tackled automatically? The issue is that a lot of domain knowledge flows into these simulations. Should the agent somehow automate that? This is, at the current state of art, impossible.
- line 502: ... phases, stages and stages. Doubling word
- Figure 14 is taken from another source. The figures should be cited not just in the text but at the figure description. This is the same for other pictures of the paper.
- the abstract consists of a list separated by semicolons and should be rewritten.
The reviewer encourages the authors to take this feedback, review the manuscript and restructure it for its comprehensiveness and completeness, and resubmit it to the journal. It would also strengthen the contribution if the authors find the time to elaborate on the gains of their example implementation, e.g., saved time during design, or discovery of more elegant solutions.
Reviewer 2 Report
Please refer to the attached file

Reviewer 3 Report
The abstract requires a better structure
The abstract requires details of some quantitative results in brief
The introduction is focused more in defining rather the acronym. It requires details of state of art and also you should indicate the authors contribution and novelty, therefore a proper formulation is required
“Materials and Methods” section is too lengthy and do not have a proper structure. From there the authors should understand the methods proposed to solve your “research theme”. Now look more a section of description of some terminology.
“Materials and Methods” requires a citation !
Not clear which is authors contribution as long as in methods you pointed out to many citations
In the section of results there should be presented only the results …now you put a lot of details which most probably belongs to methods.
A section of discussion is required
Conclusion section is lack of quantitative details
Round 2
Reviewer 2 Report
The recommendations have been addressed.
Author Response
Thanks a lot
Reviewer 3 Report
=
Author Response
Thank you for your work, the English corrections have been completed.